

**Potential effects of deep seawater discharge by an Ocean Thermal Energy**
**Conversion plant on the marine microorganisms in oligotrophic waters**
**Mélanie Giraud[1,2,3], Véronique Garçon[2], Denis de la Broise[1], Stéphane L'Helguen[1], Joël Sudre[2],**
**Marie Boye[1,4]**
[1]LEMAR - UMR 6539, IUEM Technopôle Brest-Iroise, 29280 Plouzané - France ; [2]LEGOS - UMR 5566,
31401 Toulouse cedex 9 - France ; [3]France Energies Marines, 29200 Brest - France ; [4]LOCEAN - UMR
7159, 75005 Paris - France
Corresponding author:  Dr. M. Giraud (melanie.giraud@unicaen.fr)
**Abstract**
Installation of an Ocean Thermal Energy Conversion pilot plant (OTEC) off the Caribbean coast
of Martinique is expected to use approximately 100 000 $m^3 h^{-1}$ of deep seawater for its functioning.
This study examined the potential effects of the cold nutrient-rich deep seawater discharge on the
phytoplankton community before the installation of the pilot plant. Thermal effect induced by the
deep seawater upwelled by the OTEC was described using the Regional Ocean Modeling System.
Numerical simulations of deep seawater discharge showed that a 3.0 °C temperature change,
considered as a critical threshold for temperature impact, was never reached during an annual cycle
on the top 150 m of the water column on two considered sections centered on the OTEC. The
thermal effect should be limited, less than 1 $km^2$ on the area exhibited a temperature difference of
0.3 °C (absolute value). The impact on phytoplankton of the resulting mixed deep and surface
seawater was evaluated by *in situ* microcosm experiments. Two scenario of water mix ratio (2 % and
10 % of deep water) were tested at two incubation depths (deep chlorophyll-*a* maximum: DCM and
bottom of the euphotic layer: BEL). The larger impact was obtained at DCM for the highest deep
seawater addition (10 %), with a development of diatoms, whereas 2 % addition induced only a
limited change of the phytoplankton community. This study suggested that the OTEC plant would
significantly modify the phytoplankton assemblage only in the case of a discharge affecting the DCM
and would be restricted to a local scale.
**1. Introduction**
Ocean Thermal Energy Conversion (OTEC) uses the solar energy by exploiting the temperature
gradient between surface and bottom seawater. In an OTEC plant, the cold deep seawater pumped
close to sea bottom is used to condense a working fluid (like ammonia), whereas warm surface
waters, pumped close to the surface, serve to evaporate it. The difference of pressure, generated by
the evaporation and condensation of the fluid, drives a turbine that produces mechanical energy.



This energy is then converted to electrical energy in a generator. Due to the need of a 20 °C
difference between the cold deep and the warm surface waters for the OTEC exploitation, tropical
areas are well suited for the installation of OTEC plants.
The Martinique, a tropical island of Lesser Antilles, is ideally suited for OTEC functioning with its
narrow continental slope in the Caribbean part of the island, allowing an implementation of the plant
close to the coast. The implementation of a 10 MW OTEC pilot plant off the Caribbean coast of
Martinique is expected in 2020 as part of the french NEMO project (Akuo Energy, DCNS). This
OTEC will pump approximately 100 000 $m^3.h^{-1}$ of deep seawater at 1100 m depth. In order to
optimize the energy efficiency, the deep seawater should be rejected close to the surface. However,
this large discharge could induce important disturbances on the upper ocean ecosystem, and this
impact should be estimated.
Environmental assessment of OTEC functioning was studied since the 1980's (NOAA, 1981;
2010). The deep seawater discharge was described as one of the major drivers impacting the marine
environment in OTEC plant. However, only a few studies specifically detailed this critical aspect
(Taguchi et al., 1987; Rocheleau et al., 2012). The deep seawater discharge in OTEC plant generates
a phenomenon similar to the one naturally occurring in the ocean within upwelling systems.
Equatorward winds along the coast in the eastern Atlantic and Pacific linked to atmospheric high-
pressure systems force Ekman transport and pumping, relocating coastal surface waters offshore.
Thereby, deep water transport towards the surface is generated close to the coast. In these systems,
the large amount of macronutrients and trace metals carried to the euphotic zone by the enriched
deep seawater supports a large development of the phytoplankton, making upwelling the most
productive oceanic regions (Bakun, 1990; Pauly and Christensen, 1995; Chavez and Toggweiler,
1995; Carr and Kearns, 2003). The tropical surface waters off the Caribbean coast of Martinique
exhibit low nutrients concentrations and can be significantly enriched by the deep seawater
discharge. Whereas phytoplankton assemblages in upwelling systems are usually dominated by large
phytoplankton and particularly by diatoms (Bruland et al., 2001; Van Oostende et al., 2015), the
phytoplankton community in oligotrophic systems is composed of smaller organisms (Agawin et al.,

60    2000).

Due to these important differences, it is thus of critical interest to investigate the potential effects of
the deep seawater discharge of the planned OTEC plant on the phytoplankton community off
Martinique.
In this study, the impact of deep seawater discharge on the thermal structure of surface
waters was first assessed. Modification of the surface waters stratification should indeed impact the
phytoplankton community. It is crucial to provide a depth where the deep seawater could be



discharged without significant effect on the surface layer where phytoplankton is the most abundant.
A high-resolution oceanic model was used to examine the thermal impact induced by the deep
seawater dispersion. Eight configurations of discharge depth were tested, corresponding to the
deep chlorophyll-*a* maximum (DCM), the bottom of the euphotic layer (BEL) and five depths below
the BEL. Temperature differences between numerical simulations without and with the deep
seawater discharge were compared on the upper 150 m of a vertical section.

73        The distribution of the ambient phytoplankton community and the biogeochemical

properties of the deep and surface seawater mixture that could impact the phytoplankton
community were then described. Phytoplankton distribution and assemblage were detailed in order
to assess short time and small scales variabilities of phytoplankton assemblage and primary
production in the study site.

78        Finally, in order to simulate the OTEC deep seawater input, enrichment experiments were

conducted on the future site of the pilot plant. Enrichment experiments are commonly used in
oceanography to assess the effects on phytoplankton community and primary production. For
example, large iron (Fe) enrichment experiments were conducted from 1993 to 2005 to estimate the
potential of Fe limitation on ocean primary production (De Baar et al., 2005; Boyd et al., 2007).
Several experiments also showed that macro- and micro-nutrients enrichments induce changes in the
phytoplankton community in upwelling regions (Hutchins et al., 2002) as well as in oligotrophic
regions (Kress et al., 2005). Enrichment experiments were usually conducted with mesocosms
immerged close to the surface (Escaravage et al., 1996; Duarte et al., 2000) or in laboratory under
artificial light and temperature using phytoplankton model species (Brzezinski, 1985). A laboratory
experiment intended to evaluate the effects of an OTEC seawater discharge in Hawaiian waters on
the natural phytoplankton community was previously conducted (Taguchi et al., 1987) under such
artificial conditions, and thus, it could not totally reproduce what occurred in the natural
environment. Other deep seawater discharge experiments were realized *in situ* (Aure et al., 2007;
Handå et al., 2014). For example, the use of a moored platform to upwell deep seawater and
discharge it close to the surface has shown an increase in primary production in a western
Norwegian fjord where the euphotic zone is nutrient-depleted during summer (Aure et al., 2007), as
it would be expected with the OTEC discharge. Whereas such a pumping system is well adapted for
pumping seawater at 30 m depth for example, it cannot be applied for OTEC experiments where
deep seawater must be collected far deeper (1100 m depth) and also discharged more deeply in the
water column to reduce the potential effects on the phytoplankton community. These conditions can
be obtained by the use of *in situ* microcosms, in which light and temperature are the same as in the
natural surrounding waters, avoiding additional bias, and several conditions (enrichment, incubation



depth) can be simulated. Therefore, we used the unique device of immerged microcosms we developed (Giraud et al., 2016) for assessing the effects of deep seawater discharge on the phytoplankton community. Two incubation depths (DCM and BEL) with two ratios of enriched seawater (mixtures of surface water with 2 % and 10 % of deep seawater) were tested. These experiments allowed the evaluation of critical mixing rate and discharging depth where effect was maximal.

## 2. Materials and methods

### 2.1. Modelling the thermal effect

The hydrodynamic numerical model ROMS-Regional Ocean Model System (Shchepetkin and McWilliams, 2005; 2009) was used to describe the resulting thermal effect due to OTEC functioning. The model was run in a 2-ways AGRIF configuration allowing to define a parent and child domains around the Martinique Island which are run simultaneously, transferring automatically open boundary conditions. The parent grid ranges from 63° W to 59° W and 13° N to 15.9° N with a resolution of 1/60° (around 1.8 km) while the child domain narrows the parent one and was from 61.74° W to 60.41° W and 14.21° N to 15.11° N with a resolution down to 1/180° (around 600 m). The bottom topography and coastline are interpolated from the GINA database (1/120°, www.gina.alaska.edu/data/gtopo-dem-bathymetry) (Fig. 1).

The model is forced by the monthly Climate Forecast System Reanalysis (NCEP-CFSR) for wind stress, heat and freshwater fluxes. For the open boundary conditions and initial conditions of the parent domain, a monthly climatology computed from the Simple Ocean Data Assimilation (SODA) reanalysis (Carton and Giese, 2008) was used for the dynamical variables (temperature, salinity and velocity fields). The NCEP-CFSR products do no cover the period of our mesocosm experiments (November 2013 and June 2014). The simulations were thus performed over another period when the atmospheric forcing was available. We choose the 3 years period of 1998-2000, using 1998 and 1999 as a spin-up and the last year 2000 to analyze the thermal structure and circulation field. Model outputs were stored as daily averages. The configurations were run without and with a deep seawater discharge mimicking the OTEC functioning. The deep seawater discharge was initiated on January 1$^{st}$ 2000. Eight cases of horizontal discharge settings were simulated at different depths: 1) the DCM (45 m), 2) the BEL (80 m), that were estimated on June 12$^{th}$ 2014, and 3) six depths below the euphotic zone (110 m, 140 m, 170 m, 250 m, 350 m and 500 m). In the OTEC plant, deep water will be pumped at 1100 m where temperature is around 5 °C and salinity 35. Circulation of this water through the plant system will warm it up until 8 °C prior to its release in the upper ocean. We thus



applied at the location of the OTEC plant (61°13'0'' W, 14°35'48'' N), a cold water discharge
(temperature 8 °C, salinity 35) at a flow rate of 28 m³ s⁻¹ and with a northward orientation. The
thermal impact of the cold-water source was assessed documenting the differences between
simulations without and with the modelled OTEC plant functioning over the full year 2000.

**2.2. Field observations and *in situ* experiments**


**2.2.1.   Sampling and analytical methods**


Temperature, salinity, and fluorescence profiles were performed using Seabird SBE19+ probe
with *in situ* Fluorimeter Chelsea AQUAtracka III.
Seawater was collected in the water column in ultra-clean conditions (Giraud et al., 2016) to
measure *in situ* parameters and to prepare the microcosms. Seawater and microcosms were sampled
similarly in a land laboratory a few hours after collection.
Nitrate ($NO_3^-$), nitrite ($NO_2^-$), phosphate ($PO_4^{3-}$) and silicate ($Si(OH)_4$) concentrations were
determined in filtered waters (<0.6 μm PC membrane) stored at -20 °C until analysis using a Bran +
Luebbe AAIII auto-analyzer (Aminot and Kérouel, 2007).
Filtered samples (0.2 μm; 300AC-Sartobran™ capsules) for dissolved trace metals
determination were collected under pure-$N_2$ pressure (0.7 atm) in acid cleaned low density
polyethylene bottles, acidified with ultrapure HCl (pH < 2) and stored in two plastic bags in dark at
ambient temperature. Concentrations of dissolved trace metals (cadmium, Cd; lead, Pb; iron, Fe;
zinc, Zn; manganese, Mn; cobalt, Co; nickel, Ni; and copper, Cu) were determined in UV-digested
samples by ID-ICP-MS (Milne et al., 2010) after preconcentration on a WAKO resin (Kagaya et al.,
2009) using an Element XR ICP-MS. Blanks, limits of detection, accuracy and precision (assessed
using reference samples) of the ID-ICP-MS method are reported in Table 1. The values determined
by ID-ICP-MS were in excellent agreement with the consensus values, apart for Cd that yielded
higher concentration in S-SAFE reference sample than the consensus value (Table 1).
The pH was determined using a pH ultra-electrode (pHC28) mounted on a HQ40d multi pH-
meter (HACH) with an accuracy of ± 0.002 pH unit in samples preserved with saturated $HgCl_2$ in
glass bottles hermetically closed with Apiezon grease, sealed with Parafilm® and stored in the dark
at ambient temperature.
Three complementary methods were used to analyze the phytoplankton community. Pigment
signatures were measured by HPLC (using an Agilent Technologies 1100-series) on polysulfone
filters (0.22 μm pore-size) frozen at -20 °C and stored in liquid nitrogen, after internal standard
addition (vitamin E acetate) and extraction in a 100 % methanol solution (Hooker et al., 2012). Fifty
pigments were identified and associated to phytoplankton groups (Uitz et al., 2010). Identification





and enumeration of pico-phytoplankton were realized by flow-cytometry using a BD-FACSVerse™
(Marie et al.,1999) in samples preserved in cryotube with addition of 0.25 % glutaraldehyde frozen at
-20 °C and stored in liquid nitrogen. Four groups of pico-phytoplankton were identified:
*Prochlorococcus*, picoeukaryotes (< 10 µm), and 2 groups of *Synechococcus* discriminated,
respectively, by their low and high phycourobilin (PUB) to phycoerythrobilin (PEB) ratios. Taxonomic
identification and enumeration of micro-phytoplankton (20-200 µm) and a part of nano-
phytoplankton (2-20 µm) (Dussart, 1966) were carried out using an inverted microscope (Wild M40)
in samples preserved with neutral lugol solution. Utermöhl settling chambers (Hasle, 1988) were used
for micro-phytoplankton analyses, and a smaller sedimentation chamber (2.97 mL) for the analyses of
nano-phytoplankton. When possible, phytoplankton was identified to the lowest possible taxonomic
level (species, genus or group). Biovolume of each species was also estimated from these
microscope analyses (Hillebrand et al., 1999).
### 2.2.2. *In situ* microcosm experiments
The potential impact of deep seawater discharge on the phytoplankton community was simulated
by *in situ* microcosm incubations of various deep and surface seawater mixing (Giraud et al., 2016).
The experiments were conducted from 12th (D0) to 19th (D7) of June 2014. The deep and surface
seawaters were collected at the site of the future OTEC pilot plant (61°11′52′′ W-14°37′57′′ N; Fig.
1). Microcosms bottles were incubated on two stainless steel structures set at the depths of deep
chlorophyll-*a* maximum (DCM) and at the bottom of the euphotic layer (BEL) on a mooring chain
located, for practical reasons, closer to the coast (61°10′9′′ W-14°39′8′′ N, seafloor at 220 m depth)
during 6 days (Giraud et al., 2016).
Seawater was collected at D0 at the depths of DCM (45 m depth) and BEL (80 m depth)
identified on the future OTEC site from the fluorescence profile, and close to the bottom (1100 m
depth corresponding to the pumping depth of the future OTEC plant) in ultra-clean conditions.
Deep seawater was mixed in three proportions (0 % as a control hereafter referred to as "Control", 2
% as a low input called "2 % of deep seawater", and 10 % as a large input called "10 % of deep
seawater") with DCM and BEL waters. Each resulting mixture was distributed in 2.3 L polycarbonate
bottles filled up to overflow level, of which four replicates per mixing condition per depth were
immersed at their respective sampling-depth for 6 days; duplicates per mixing condition per depth
were kept in dark at 25 °C for a few hours until sampling for later characterization of phytoplankton
assemblage and biogeochemical properties at D0 (called "Surrounding waters D0"); and duplicates
per mixing condition per depth were used to estimate carbon and $NO_3^-$ uptakes at D0 (called
"Surrounding waters D0") as described below.



Same sampling and mixtures were realized at day 6 (D6, June 18th) just to evaluate the temporal
evolution in the natural environment, resting on duplicate bottles per mixing condition per depth for
phytoplankton and biogeochemical characterizations at D6 (called "Surrounding waters D6") and
using other duplicates to estimate carbon and $NO_3^-$ uptakes at D6 (called "Surrounding waters D6").
After the 6 days incubation, all the incubated microcosm bottles on the mooring (called
"Microcosm D6") were brought on board. A quarter of each four replicates per condition was put in
a new 2.3 L clean bottle and used to estimate carbon and $NO_3^-$ uptakes after 6 days of incubation
(called "Microcosm D6"). The remaining microcosm contents were kept for sampling and analysis.

### 2.2.3.   Carbon and nitrate uptakes

Carbon (primary production) and $NO_3^-$ uptake rates were estimated in the same sample using the
dual $^{13}C/^{15}N$ isotopic label technique (Slawyk et al., 1977). Immediately after sampling, $^{13}C$ tracer
($NaH^{13}CO_3$, 99 atom%, Eurisotop, 0.25 mmol$^{13}C$ mL$^{-1}$) and $^{15}N$ tracer ($Na^{15}NO_3$, 99 atom%, Eurisotop,
1 μmol$^{15}N$ mL$^{-1}$) were added to seawater mixtures at $10^{-3}$:1 v/v ratio. The initial enrichment was 10
atom% excess of $^{13}C$ for the bicarbonate pool and 16-95 atom% excess of $^{15}N$ for the $NO_3^-$ pool
depending on the ambient $NO_3^-$ concentration. The $^{13}C/^{15}N$ amended bottles were incubated for 24
h on the mooring line at the DCM and BEL depths, after which 1 L samples were filtered onto pre-
combusted (450 °C, 4 h) glass fiber filters (Whatman). Filters were stored at -20 °C and oven dried
(60 °C, 24 h) prior to analysis. Concentrations of carbon (POC), nitrogen (PON) as well as $^{13}C$ and $^{15}N$
enrichments in particulate matter were measured with a mass spectrometer (Delta plus,
ThermoFisher Scientific) coupled to a C/N analyzer (Flash EA, ThermoFisher Scientific). Standard
deviations were 0.009 μM and 0.004 μM for POC and PON, and 0.0002 atom% and 0.0001 atom%
for $^{13}C$- and $^{15}N$-enrichments, respectively.
The absolute uptake rates (ρ, in μmol L$^{-1}$ h$^{-1}$) were calculated for nitrogen (Dugdale and Wilkerson,
1986) and carbon (Fernández et al., 2005) using the particulate organic concentrations measured
after 24 h of incubation. These rates were converted into biomass specific uptake rates (V, in μmol
μmol POC or PON$^{-1}$ h$^{-1}$) by dividing ρ by POC or PON. The addition of $^{15}N$ tracer would cause a
substantial increase in dissolved inorganic nitrogen concentrations especially in the surface waters
and, in turn, an overestimation of uptake rates (Dugdale and Wilkerson, 1986; Harrison et al., 1996).
The $NO_3^-$ uptake rates were corrected for this perturbation (Dugdale and Wilkerson, 1986) using a
half-saturation constant of 0.05 μmol.L$^{-1}$ characteristic for nitrogen-poor oceanic waters (Harrison et
al., 1996) and the measured $NO_3^-$ concentration. Overestimation was low (< 5 %) in samples with an
addition of deep seawater but it was of about 50 % in samples without deep seawater addition. The
uptake rates measured in these samples represented therefore estimations rather than actual values.





### 2.2.4. Statistical analyses

Kruskal-Wallis test was applied on the set of pigments concentrations, pico-phytoplankton
abundances and macronutrients concentrations. If significant differences ($p < 0.05$) were found,
Mann-Whitney test was run to identify the samples significantly different. Statistical analyses were
performed using Statgraphics Centurion XVI software.

## 3. Results

### 3.1. Modeling of the deep seawater discharge

#### 3.1.1. Model evaluation

We compared modeled daily profiles (temperature, salinity) of June and November 2000 with *in*
*situ* CTD data at OTEC station we recorded in June 2014 and November 2013 (Fig. 2 a-b).
In June, the modeled and observed vertical profiles of temperature were quite in agreement with
a well mimicked thermocline depth. However, a warm bias of ~1.5 °C was simulated by the model in
the top 50 m. Between 300 and 500 m depth, a cold bias of ~1.5 °C depth was also observed. The
modeled and experimental salinity profiles presented a similar pattern. However, the salinity was
largely overestimated by the model in the top-120 m, especially in the upper 60 m (by ~2 units), as
compared to field observations. Between 120 m and 150 m, the model slightly underestimated the
salinity.
In November, the thermocline and halocline were well reproduced with modeled vertical profiles
of temperature and salinity, in good agreement with observations. However, temperature was
slightly overestimated by the model, with warm bias of ~0.8 °C. At deeper depths, the modeled and
observed temperatures were in excellent agreement. Salinity was underestimated by the model
within the top 50 m by ~1 unit, and between 70 and 200 m depths by at maximum 0.3 units. Below
200 m depth, modeled and observed salinities exhibited similar profiles.
ADCP measurements (horizontal velocity and direction of currents) were made by our DCNS
partner at the study site for a feasibility study, but in June 2011 (between 40 m and 800 m depths).
ADCP data were compared to model outputs for June 2000 (Fig. 3). Current directions were quite
similar between model outputs and ADCP data with a mean direction toward the South/South-East.
The horizontal velocity norm was also quite close between both data sets with larger velocity close
to the surface at ~50 m depth. Larger difference appeared in subsurface in ADCP data but similar
trends were observed and values were relatively close.
Modeled physical properties (temperature, salinity, currents) were therefore quite similar to those
directly observed at the study site. The small differences observed between model and field data are



likely due to inter-annual variability since years examined were indeed different for the model
simulation (2000) and the field data (2011, 2013 and 2014).

### 3.1.2. Impact of the deep seawater discharge on the thermal structure in surface

In order to assess the deep seawater discharge impact on the thermal structure of the upper 150
m of the water column, the dispersion of temperature differences (ΔT in °C) obtained without and
with the deep seawater discharge in the model outputs was examined on two vertical sections. A
section of 124 km for the large domain (corresponding to the child domain) and another section of
10 km for the near-OTEC domain (defined from 61.24° W to 61.17° W and 14.60° N to 14.67° N)
were defined, both centered on the OTEC site and parallel to the coast (Fig. 1). Presently, there are
no environmental standards defining threshold levels for temperature difference that will be induced
by an OTEC deep seawater discharge. So, the study relied on the World Bank Group prescriptions
for liquefied natural gas facilities which set at 3 °C the temperature difference limit at the edges of
the zone where initial mixing and dilution take place (IFC, 2007).
We thus considered for each discharge depth the cooling and warming outputs from the model,
which exhibit a $|\Delta T| \geq 3$ °C. Areas (in % of the considered domain) impacted by these cooling and
warming effects were added (absolute values) in order to compare the potential impact of each
discharge depth configuration. None of the discharge depth configurations could produce a
modification of the thermal structure of the top 150 m of the water column, higher than or equal to
the considered temperature threshold ($|\Delta T| \geq 3$ °C), for both domains sections.
Then, a lower temperature difference of 0.3 °C (absolute value) was considered. This temperature
difference represented a low threshold as compared to the World Bank Group prescriptions (IFC,
2007) that instead represent a high threshold. The areas exhibiting a $|\Delta T| \geq 0.3$ °C in the top 150 m
(Table 2) were extremely small (< 1 km²) and were not significantly different in both sections and at
the different discharge depths, on an annual average and in June (our experimental period).

### 3.2. Biogeochemical properties and phytoplankton community

### 3.2.1. Expected biogeochemical properties of the resulting mixed waters

The pH was very similar at the DCM and BEL at the OTEC site on D6 (8.24 and 8.25,
respectively), whereas deep seawater-pH showed lower value (7.81). The addition of 2 % and 10 %
deep seawater to surface waters could thus induce a pH-decrease of respectively, 0.01 and 0.07 unit.
Hence, the effect on pH could be rather limited compared to the 0.1 pH decrease (from 8.2 to 8.1)
between the pre-industrial time and the 1990's [39].



$NO_3^-$ and $PO_4^{3-}$ concentrations (Table 3) were below the detection limit (< 0.02 μM) at the DCM
(55 m) and BEL (80 m) at the OTEC site on observational D4 (June 16th 2014), whereas $Si(OH)_4$
concentrations were above detection limit (> 0.08 μM), particularly at the DCM (2.4 μM). $NO_2^-$
concentrations showed the highest values at the BEL whereas they were negligible at the DCM
(<0.02 μM). In deep seawater, as commonly observed, $NO_3^-$, $PO_4^{3-}$ and $Si(OH)_4$ concentrations were
largely higher compared to the surface (Table 3). The 2 % and 10 % deep water additions
represented a large input for $NO_3^-$ (from <0.02 μM to 0.54 and 2.71 μM, respectively). If the 10 %
ratio also induced a large input of $PO_4^{3-}$ (from <0.02 to 0.19 μM), the input of 2 % deep water was
more limited (0.04 μM). The effect of 2 % or 10 % deep seawater addition was more limited for
$Si(OH)_4$ relatively to $NO_3^-$ and $PO_4^{3-}$ input, yet it accounted for 50-63 % increase for 10 % deep
seawater addition (Table 3). Finally, because deep and DCM waters were $NO_2^-$ depleted, the deep
seawater input did not modify the $NO_2^-$ concentration at the DCM. At the BEL, $NO_2^-$ concentration
was higher and the 10 % addition slightly diluted $NO_2^-$ at this depth.
Mn showed maximum concentrations in the surface layer on D4 at the OTEC site (Table 4)
decreasing with depth as observed close to the Lesser Antilles in the Atlantic Ocean (Mawji et al.,
2015), but the measured surface concentrations were particularly high, especially at the DCM. Fe
that commonly dispatches hybrid distribution combining a nutrient-type profile in surface waters and
a scavenged-type distribution in deep waters (Bruland, 2003) also exhibited high surface values,
particularly at the DCM (Table 4). Cd, Zn, Co, Ni, and Cu dispatched nutrient-type profiles, whereas
Pb exhibited scavenged-type profile (Nozaki, 1997; Gruber, 2008), but like for dissolved Fe and Mn,
their concentrations in the upper waters were particularly high (Table 4). For all trace metals at both
depths, the 2 % deep seawater addition will not induce significant changes in their surface
concentrations (Table 4). The 10 % deep seawater addition could increase Cd, Ni and Zn
concentrations in surface waters (Table 4), whereas it would not constitute an input of Pb, Cu, Co,
and Fe, and it can even dilute Mn (Table 4).
The surface waters can thus be enriched in macronutrients ($NO_3^-$, $PO_4^{3-}$) when submitted to a
deep seawater discharge (particularly with 10 % deep seawater addition) in proportion depending
on the depth. The same scheme can be applied in some of the dissolved trace metals (Cd, Ni, Zn)
when a large ratio of deep seawater (10 %) is discharged.
### 3.2.2. Phytoplankton community in the natural environment
A set of seven accessory pigments identified as biomarkers of specific taxa (Uitz et al., 2010;
Table 5) were analyzed at OTEC station at D0, D4 and D6 in surrounding surface waters (Fig. 4), as
well as population abundance and their biovolume using light microscopy (Fig. 5).



The total chlorophyll a (TChl a defined as the sum of chlorophyll a and divinyl chlorophyll a), a proxy of the phytoplankton biomass, was higher at DCM than at BEL, as usually observed, by about two-folds. The fucoxanthin (biomarker of diatoms) concentrations were similar at the DCM and BEL on D0 (Fig. 4), like the total abundance of diatoms (Fig. 5). Fucoxanthin concentration increased by D4 and then by D6 at the DCM, corresponding to increases of cumulated diatoms biovolume on D4 (Fig. 5) and of diatoms abundance on D6 (Fig. 5). Peridinin, a biomarker of dinoflagellates, was detected at the DCM unlike at the BEL, with relatively high abundance and biovolume of dinoflagellates (Fig. 5). The 19'-hexanoyloxyfucoxanthin (biomarker of haptophytes) concentration (Fig. 4) and the prymmnesiophytes (haptophyte) abundance and biovolume (Fig. 5) showed higher values at the DCM than at the BEL only at D4.

At the DCM, dinoflagellates largely dominated the nano- and micro-phytoplankton assemblage with the largest abundance and biovolume. Whereas prymnesiophytes showed the second highest abundance, its biovolume was very low, on the contrary to diatoms that dispatched lower abundance but higher biovolume (Fig. 5). At the BEL, dinoflagellates, prymnesiophytes and diatoms showed similar abundance, dinoflagellates and the diatoms occupied the major part of the total biovolume. Three groups of dinoflagellates were observed by light microscopy but they could not be identified at species level. However, their small size and the lack of colored starch (using lugol) in the cytoplasm suggested they were mixotrophic or heterotrophic population. Furthermore, the low concentrations of peridinin in samples supported this assumption.

At both depths, light microscopy analyses suggested that the large cyanobacteria, mainly Trichodesmium sp., were low in abundance and biovolume. Flow cytometry identification and count indicated that the small cyanobacteria Prochlorococcus dominated the pico-phytoplankton assemblage, but they showed a significant decrease from D0 to D6 (Fig. 6). A significant portion of Synechococcus was also observed while picoeukaryotes were poorly represented. Both Prochlorococcus and Synechococcus showed higher abundance at the DCM than at the BEL (by 65 % and 86 %, respectively), in line with the pigments analyses of zeaxanthin (biomarker of cyanobacteria) and total chlorophyll b concentrations (prochlorophytes).

### 3.2.3. Primary production and nitrate uptake in the natural environment

The phytoplankton distribution and assemblage can partly drive the intensity of primary production, so the specific uptake rate of carbon ($V_C$; Fig. 7) and $NO_3^-$ ($V_{NO3^-}$) were estimated at D0 and D6.




$V_C$ in surrounding surface waters was relatively low at D0 (Fig. 7) indicating low primary
production in these poor-nutrients waters. Yet, $V_C$ was approximately four-times higher at the DCM
($2.10^{-3}$ $h^{-1}$) than at the BEL ($5.10^{-4}$ $h^{-1}$) at D0, but drastically decreasing on D6 at the DCM (to ~$6.10^{-4}$
$h^{-1}$). $V_{NO3-}$ were also very low at D0 ($1.10^{-3}$ $h^{-1}$ at DCM, $4.10^{-3}$ $h^{-1}$ at BEL) and drastically decreased at
D6, below the detection limit (data not shown).
**3.3. Impacts on the phytoplankton community of the deep seawater discharge**
**3.3.1.  Changes in the phytoplankton assemblage**
At the DCM, TChl *a* was similar in all treatments ($p < 0.05$) after 6 days of incubation in
microcosms (Fig. 8). Only fucoxanthin and 19'-butanoyloxyfucoxanthin showed significant ($p < 0.05$)
higher concentrations in 10 % enrichments as compared to controls, indicating higher abundance
and/or biovolume of diatoms and haptophytes. The other diagnostic pigments did not show any
significant difference between enriched microcosms and controls. Picoeukaryotes and
*Synechococcus* abundances did not show significant variations between the treatments (Fig. 9a).
Reversely, *Prochlorococcus* population showed higher ($p < 0.05$) abundance both in 2 % and 10 %
enriched microcosms as compared to controls (Fig. 9a).
At the BEL, after the 6 days incubation period, pigments concentrations were below the
detection limit indicating very low abundance of phytoplankton. Pico-phytoplankton did not show
significant variations between the treatments and the controls (Fig. 9b). Pico-phytoplankton were
clearly much less abundant at the BEL (< 1000 cells $mL^{-1}$) than at DCM (Fig. 9b), 20-times even lower
than that observed in surrounding waters at this depth on D6. For comparison, total abundance at
the DCM was ~5-times lower in incubated microcosms on D6 compared to surrounding surface
waters.
**3.3.2.  Changes in the primary production and nitrate uptake**
Deep water inputs (2 % and 10 %) to surrounding waters collected at the DCM on D0 led to
an increase of $V_C$ within 24 h compared to the controls (by 43 % and 48 %, respectively; Fig. 7); but
they had no effect on D6 despite very low value in natural waters at this depth ($6.10^{-4}$ $h^{-1}$). The 6 days
incubated microcosms showed very low $V_C$ in all treatments (Fig.7). At the BEL, $V_C$ were quite similar
on D0 and D6 and after 6 days of incubation, without significant differences between the treatments
(Fig. 7). $V_{NO3-}$ measured in microcosms after a 6-days *in situ* incubation were below the detection
limit (data not shown).






### 4. Discussion

#### 4.1. Natural variabilities in the oligotrophic area

##### 4.1.1. Modeling of the deep seawater discharge

Salinity field data showed large seasonal variations, with low values in June 2014 (34.6 on the top 50 m) and much higher values in November 2013 (35.5 on the top 50 m). The model run for year 2000 did not fully reproduce these variations. Indeed, salinity was overestimated by the model in June whereas it was underestimated in November. The observations we made at the OTEC station showed that the low salinity observed in June was associated with high $Si(OH)_4$ concentrations. High $Si(OH)_4$ levels in fresher seawater have been already reported in surface waters in the Caribbean Sea and they were attributed to Amazon and Orinoco fresh rivers inputs (Steven and Brooks, 1972; Moore et al., 1986; Muller-Karger et al., 1995; Hu et al., 2004). Fresh surface waters enriched in $Si(OH)_4$ (Moore et al., 1986; Edmond et al., 1981) can be transported from the Amazon and Orinoco rivers towards the Caribbean Sea by the North Brazil Current and the Guiana Current (Muller-Karger et al., 1988, 1995; Osborne et al., 2014, 2015). It is likely that the rivers discharges and thus its inputs in the Caribbean Sea were quite different between 2000 (modeled year) and 2014 (*in situ* observations), thus explaining the discrepancy between modeled and observed salinities. Meso- and submeso-scale features resulting from the rivers flows could also induce short-term variability in the area and then could explain the observed differences.

##### 4.1.2. Biogeochemistry and phytoplankton community structure

The very low $PO_4^{3-}$ and $NO_3^-$ concentrations recorded in the oligotrophic surrounding surface waters were likely favorable to the development of small phytoplankton, especially to the cyanobacteria as shown with the significant occurrence of *Prochlorococcus* in these waters, which are typical of poor nutrient waters (Partensky et al., 1999). In line with the very low $V_{NO3^-}$ measured here, it has been shown that $V_{NO3^-}$ by *Prochlorococcus* represents indeed only 5-10 % of its nitrogen uptake whereas reduced nitrogen substrates ($NO_2^-$, ammonium, and urea) uptake accounts for more than 90-95 % (Casey et al., 2007). By contrast, the development of larger phytoplankton taxa (particularly diatoms), which have higher $NO_3^-$ and $PO_4^{3-}$ requirements for their growth, were probably limited by these elements. Actually, $NO_3^-$ and $PO_4^{3-}$ concentrations in surrounding waters at the DCM were both lower than the detection limit (< 0.02 µM at D4) which is much lower than the average values of half-saturation constants for diatoms (1.6 ± 1.9 µM for $NO_3^-$ and 0.24 ± 0.29 µM for $PO_4^{3-}$; Sarthou et al., 2005). For $Si(OH)_4$, surrounding surface concentrations at DCM (2.39 µM) were in the range of diatoms half-saturation constants (3.9 ± 5.0 µM; Sarthou et al., 2005), hence the



diatoms development was probably not limited by $Si(OH)_4$. Furthermore, diatoms showed low
abundance in spite of relatively high $Si(OH)_4$ and dissolved trace metals (in particular Fe)
concentrations in surface waters. The potential of Fe limitation on phytoplankton community has
been reported previously in upwelling systems, with an apparent half-saturation constant for diatoms
growth of 0.26 nM Fe in the Peru Upwelling system (Hutchins et al., 2002). This constant is far lower
than the concentration of Fe measured in surrounding waters at DCM (1.08 ± 0.03 µM at D4),
suggesting that diatoms were probably not limited by Fe. This further supports growth limitation of
diatoms by $NO_3^-$ and/or $PO_4^{3-}$.
Advection of waters from Amazon and Orinoco rivers can explain the relatively high $Si(OH)_4$
observed in the Caribbean Sea. However, little information is available on the input of trace metals
by these waters into the Caribbean Sea. Amazon river can be a source of dissolved Fe, Cu, Ni, Pb
and Co for the western-subtropical North Atlantic (Tovar-Sanchez and Sañudo-Wilhelmy, 2011), but
this input can decrease rapidly away from its source like it has been shown for Co in the Western
Atlantic (Dulaquais et al., 2014). Those inputs into the Caribbean Sea will have to be further
examined, especially for Fe, Cd, Ni, Zn, Mn whose relatively high concentrations were detected in
the $Si(OH)_4$-enriched surface waters of this study. Additionally, other inputs of trace metals such as
atmospheric deposition can also increase surface concentrations, and those inputs can be substantial
(Shelley et al., 2012).

### 443     4.1.3. Primary production

Despite low $V_C$ on D0 and D6 at the DCM, primary production still indicated much higher value
on D0 compared to D6 that was associated with higher TChl $a$ (Fig. 4a). The decrease of divinyl-
chlorophyll $a$ (*Prochlorococcus*) concentration [58] over the 6 days of observation can account for the
decrease of TChl $a$, whereas chlorophyll $a$ concentrations did not vary significantly during this period.
The *Prochlorococcus* abundance was also lower by two-times on D6 compared to D0 (Fig. 6a). On
the contrary, fucoxanthin (diatoms) increased by four-times over the 6 days (Fig. 4 a), as well as the
diatoms abundance (by three-times; Fig. 5a). In turn, the increase in diatoms abundance was not
associated with an increase in primary production. Instead, the observed decrease in primary
production can be due to the decrease in *Prochlorococcus* abundance. In tropical and subtropical
waters, pico-phytoplankton can indeed contribute to more than 80 % of the primary production
(Platt et al., 1983; Goericke and Welschmeyer, 1993). The development of diatoms population likely
did not compensate the large decrease in *Prochlorococcus* abundance (from 141 to 63 $10^3$ cells mL$^-$
$^1$).





### 4.2. Impact of deep seawater discharge

#### 4.2.1. Temperature effects

The numerical simulation showed that the area impacted in the top-150 m by a temperature difference larger than or equal to 0.3 °C (absolute value) was lower than 1 km² (~2-3 % of the considered domain) and was insensitive to the injection depth or to the size of the tested domain (Table 2). This suggests that temperature difference might rather be linked to internal variability of the system. Since the effect of the discharge appears undetectable within 2-3 % variation of the model, it can be deduced that in a worst-case scenario, only 3 % of the small domain (300 m along the section, down to the 150 m depth) would be impacted by a temperature difference larger than or equal to 0.3 °C (absolute value). The impact of a 0.3 °C temperature variation on the growth of diatoms, notably on *Pseudonitzschia pseudodelicatissima* species that were observed in our study area, is limited to a change in the growth rate of 0.03 d$^{-1}$ [61]. For *Synechococcus,* a 0.3 °C variation of the temperature would also have a limited impact on the growth, with a variation of only 0.02 d$^{-1}$ (Boyd et al., 2013), like for *Emiliania huxleyi* (coccolithophyceae) for which the induced variation of maximum growth rate will be lower than 0.01 d$^{-1}$ (Fielding, 2013). The thermal effect on the phytoplankton assemblage could thus be considered negligible.

#### 4.2.2. Impact on the phytoplankton community

Microcosms enrichment of DCM waters with 10 % of deep seawater led after 6 days to a significant increase ($p < 0.05$) of fucoxanthin (diatoms) and 19'-butanoyloxyfucoxanthin (haptophytes) by 71 % and 77 %, respectively, as compared to the controls. If the 2 % enrichment also showed similar trends, the differences of diagnostic pigments concentrations were not significant. $NO_3^-$ and $PO_4^{3-}$ concentrations induced by 10 % deep-water input on D0 (2.57 ± 0.13 µM and 0.14 ± 0.2 µM, respectively; Giraud et al., 2016) were close to $NO_3^-$ and $PO_4^{3-}$ half-saturation constants of diatoms (1.6 ± 1.9 µM and 0.24 ± 0.29 µM, respectively; Sarthou et al., 2005). The 10 % enrichment could thus support a development of diatoms. On the contrary, $NO_3^-$ and $PO_4^{3-}$ enrichments induced by 2 % addition of deep-water were too low (0.57 ± 0.02 µM and 0.04 ± 0.00 µM, respectively; Giraud et al., 2016) compared to these half-saturation constants to support the diatoms development. Therefore, the diagnostic pigments suggested a significant response proportionally to the amount of added deep seawater.

*Prochlorococcus* were also more abundant ($p < 0.05$) in 2 % and 10 % treatments as compared to the controls. This lack of further *Prochlorococcus* population increase in 10 % treatments could be

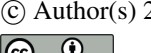


attributed to a higher grazing pressure by haptophytes and/or to $NO_3^-$ and $PO_4^{3-}$ too rich conditions
(Giraud et al., 2016).
Phytoplankton assemblage widely evolved in surrounding waters, from a predominance of pico-
phytoplankton (*Prochlorococcus*) on D0 towards a higher abundance of micro-phytoplankton
(diatoms) on D6. In order to assess if the impact on the phytoplankton assemblage due to 10 %
deep seawater addition (with a shift towards the diatoms) was in the range of the natural variation
observed in the surrounding surface waters, 10 % deep seawater microcosms phytoplankton
assemblage was compared to the natural phytoplankton assemblage.
Whereas microcosm controls showed a lower *Prochlorococcus* abundance (Fig. 9a) than
surrounding surface waters on D6 ($p < 0.05$), the 10 % microcosms additionally showed, higher
fucoxanthin (diatoms) and 19'-butanoyloxyfucoxanthin (haptophytes) by about 142 % and 317 %
(Fig. 8), respectively, as compared to natural waters at D6. Furthermore, 10 % enrichments showed a
fucoxanthin increase over the 6 days period by 3-times higher than in surrounding waters, whereas
controls only showed an increase by 1.5-times higher than in surrounding waters. Therefore, it can
be concluded that the 10 % deep seawater enrichment induced higher variations of the
phytoplankton assemblage than those observed from D0 to D6 in surrounding surface waters.
$V_C$ were higher ($p < 0.05$) both in 2 % and 10 % enrichments on D0 as compared to controls,
suggesting a positive response of phytoplankton to the deep seawater addition. Conversely, there
was no carbon-uptake rate difference ($p < 0.05$) between controls and enriched waters (with 2 % and
10 % of deep seawater) at D6 with the 6 days incubated microcosms, suggesting that the observed
community modifications did not change the primary production. Indeed, the phytoplankton
community was quite similar in surrounding surface waters on D6 and in 6 days-incubated microcosm
controls. Thus, only the initial phytoplankton assemblage and initial primary production in
surrounding surface waters would influence the response of the phytoplankton community and its
production.
At the BEL, after 6 days of incubation, deep seawater addition experiments clearly showed lower
effects on the phytoplankton community than at the DCM. Indeed, whereas significant differences
($p < 0.05$) between 10 % enrichments and controls were observed in diagnostic pigments
concentrations at the DCM, pigments concentrations were too low at the BEL to be quantified. It
can be suggested that the lower population and lower carbon uptake could be related to the lowest
light availability.
Overall, the phytoplankton response was proportional to the amount of added deep seawater. If
the phytoplankton assemblage significantly varied over time in the environment, the 10 % deep
seawater enrichment showed larger variations (for diatoms and haptophytes) than those observed in



the natural environment. The DCM should be more impacted than the BEL by the deep seawater
discharge even with a large deep seawater input. On the other hand, the impact on the primary
production largely depended on the initial phytoplankton assemblage, which was quite variable over
time. The modification of the phytoplankton community to a deep seawater input could also be
depending on the initial phytoplankton community. For that, the microcosm experiments did not
allow drawing a scenario over the long term of the potential modifications of the primary production
and the phytoplankton community associated to the deep seawater discharge by an OTEC.
Light microscopy analyses showed a large abundance of dinoflagellates at the DCM (between
9,240 and 20,400 cells mL$^{-1}$ on D6 and D4; Fig. 5 a) which could be mixotrophic or heterotrophic and
thus probably exert a grazing pressure on the phytoplankton, particularly on the pico-phytoplankton
(Liu et al., 2002). However, in this study, the zooplankton larger than 200 µm and its potential control
on the phytoplankton community were not considered and should be examined in future studies.

## 5. Conclusion

Two complementary approaches were applied to study the potential effects of the deep
seawater discharge of the planned OTEC plant on the phytoplankton community off Martinique.
Because the distribution and the development of phytoplankton are directly linked to the surface
stratification, it is important to assess the thermal effect of deep seawater by an OTEC plant.
Modelling of the deep seawater discharge showed that the thermal structure of the top 150 m of the
water column on large and near-OTEC sections should be very slightly impacted for the lowest
considered temperature differences $|\Delta T| \geq 0.3$ °C. If World Bank Group prescriptions of not
exceeding a higher temperature difference of 3 °C are followed, the environmental perturbations
potentially caused by the operation of the OTEC should be considered negligible. The area where
the 150 m-depth waters are impacted by the lowest considered temperature differences $|\Delta T| \geq 0.3$
°C would not exceed 1 km$^2$ in a worst-case scenario.
The phytoplankton community and its production could be impacted by a large deep seawater
input. Whereas pico-phytoplankton currently largely dominates the phytoplankton assemblage, a
ratio of 10 % of deep seawater in DCM waters could induce a shift toward the diatoms and micro-
phytoplankton. The ratio of 2 % of deep seawater in DCM waters only showed significant higher
*Prochlorococcus* abundance than controls, but the assemblage and the primary production were not
modified by this lower input. The stimulation of *Prochlorococcus* could be due to one or some of the
following causes: $NO_3^-$ and/or $PO_4^{3-}$ supply, trace metal supply, lowered pH (higher availability of
dissolved inorganic carbon).



Although significant, these results would have to be extended to larger temporal scale, and the
phytoplankton interactions with higher trophic levels (such as zooplankton) must be studied.
Because no environment standards on the deep seawater discharge effects are available yet, a
rigorous monitoring of the phytoplankton community, biogeochemical parameters distribution and
of the water column stratification must be established as soon as the OTEC is implemented and
during its continuous functioning.
**Acknowledgements**
This work was supported by France Energies Marines and part of the IMPALA project. We would like
to thank the Captains and crew members of the "Pointe d'Enfer", and the scientists in the laboratory
at the University of the French West Indies and Guiana at Martinique; Dominique Marie (UPMC,
Roscoff, France) and Christophe Lambert (LEMAR, France) for their help with the flow cytometry,
and Anne Donval (LEMAR, France) for the pigment analyses.






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





**Tables**

**Table 1-** Comparison of analyses of SAFe (Sampling and Analysis of iron) S and D2 reference samples (http://www.geotraces.org/science/intercalibration) between ID-ICPMS values (this study) and the consensus values. Our mean reagent blanks (based on all blank determinations) for dissolved Cd, Pb, Fe, Ni, Cu, Zn, Mn and Co, and detection limits of ID-ICPMS estimated as three times the standard deviation of the mean reagent blanks are also shown.

|  | Cd (pM) | Pb (pM) | Fe (nM) | Ni (nM) | Cu (nM) | Zn (nM) | Mn (nM) | Co (pM) |
|---|---|---|---|---|---|---|---|---|
| **SAFe D2** |  |  |  |  |  |  |  |  |
| This study | 948.83 ± 65.95 | 28.86 ± 4.44 | 0.898 ± 0.098 | 8.60 ± 0.36 | 2.15 ± 0.16 | 7.29 ± 0.27 | 0.40 ± 0.05 | 40.12 ± 3.88 |
| Consensus values | 986.00 ± 23.00 | 27.70 ± 1.50 | 0.933 ± 0.023 | 8.63 ± 0.25 | 2.28 ± 0.15 | 7.43 ± 0.25 | 0.35 ± 0.05 | 45.70 ± 2.90 |
| n= | 20 | 20 | 18 | 19 | 22 | 13 | 23 | 23 |
| **SAFe S** |  |  |  |  |  |  |  |  |
| This study | 7.24 ± 1.57 | 48.42 ± 6.08 | 0.087 ± 0.025 | 2.56 ± 0.55 | 0.55 ± 0.06 | 0.07 ± 0.06 | 0.75 ± 0.05 | 2.85 ± 0.81 |
| Consensus values | 1.10 ± 0.30 | 48.00 ± 2.20 | 0.093 ± 0.008 | 2.28 ± 0.09 | 0.52 ± 0.05 | 0.07 ± 0.01 | 0.79 ± 0.06 | 4.80 ± 1.20 |
| n= | 25 | 27 | 15 | 25 | 30 | 10 | 27 | 28 |
| **Detection Limit** | 0.996 | 0.613 | 0.032 | 0.096 | 0.011 | 0.129 | 0.001 | 0.07 |
| **Blanks** | 0.716 | 1.809 | 0.061 | 0.040 | 0.019 | 0.129 | 0.003 | 0.32 |

**Table 2-** Area (km$^2$) impacted in the top-150 m by a temperature difference $|\Delta T| \geq 0.3$ °C on two vertical sections centered on the OTEC, considering eight depths of deep seawater discharge (45, 80, 110, 140, 170, 250, 350, 500 m), average and root mean square for the year 2000 (from the monthly data) and for June 2000.

| Depth of deep water discharge | Mean Year 2000 | | June 2000 | |
|---|---|---|---|---|
|  | Large domain | Near-OTEC domain | Large domain | Near-OTEC domain |
| **45 m** | 0.4 ± 0.4 | 0.0 ± 0.1 | 0.0 | 0.0 |
| **80 m** | 0.6 ± 0.7 | 0.1 ± 0.1 | 0.4 | 0.0 |
| **110 m** | 0.6 ± 0.5 | 0.0 ± 0.1 | 0.9 | 0.0 |
| **140 m** | 0.4 ± 0.5 | 0.1 ± 0.1 | 0.1 | 0.0 |
| **170 m** | 0.5 ± 0.8 | 0.0 ± 0.1 | 0.5 | 0.0 |
| **250 m** | 0.5 ± 0.7 | 0.1 ± 0.1 | 0.1 | 0.0 |
| **350 m** | 0.5 ± 0.5 | 0.1 ± 0.1 | 0.0 | 0.0 |
| **500 m** | 0.5 ± 0.5 | 0.1 ± 0.1 | 0.3 | 0.0 |




**Table 3-** Nitrate, silicate, phosphate and nitrite concentrations on June 16th 2014 (D4) at the deep
chlorophyll maximum (DCM), at the bottom of the euphotic layer (BEL), and at the deep seawater
pumping depth. Concentrations were measured at the OTEC site (0 % addition of deep waters) and
calculated for 2 % and 10 % deep seawater additions.

| Depth (m) | Deep seawater ratio | $[NO_3^-]$ ($\mu$M) | $[Si(OH)_4]$ ($\mu$M) | $[PO_4^{3-}]$ ($\mu$M) | $[NO_2^-]$ ($\mu$M) |
|---|---|---|---|---|---|
| DCM | 0 % | < 0.02 | 2.39 | < 0.02 | 0.02 |
| | 2 % | 0.54 | 2.88 | 0.04 | 0.02 |
| | 10 % | 2.71 | 4.82 | 0.19 | 0.02 |
| BEL | 0 % | < 0.02 | 1.46 | < 0.02 | 0.32 |
| | 2 % | 0.54 | 1.96 | 0.04 | 0.32 |
| | 10 % | 2.71 | 3.98 | 0.19 | 0.29 |
| 1100 | 100 % | 27.11 | 26.69 | 1.87 | <0.02 |



**Table 4-** Concentrations of dissolved trace metals (in nM): Mn, Fe, Cd, Zn, Co, Ni, Cu, Pb measured
on June 16th 2014 (D4) at the OTEC site at the DCM, BEL and 1100 m (0 % addition of deep waters),
and their calculated concentrations in the mixtures with 2 % and 10 % addition of deep water.

| Depth (m) | Deep seawater ratio | Mn (nM) | Fe (nM) | Cd (nM) | Zn (nM) | Co (nM) | Ni (nM) | Cu (nM) | Pb (nM) |
|---|---|---|---|---|---|---|---|---|---|
| DCM | 0 % | 2.97 ± 0.17 | 1.08 ± 0.03 | 0.03 ± 0.01 | 1.54 ± 0.04 | 0.05 ± 0.00 | 2.22 ± 0.10 | 1.70 ± 0.18 | 0.03 ± 0.00 |
| | 2 % | 2.92 | 1.08 | 0.04 | 1.56 | 0.05 | 2.29 | 1.70 | 0.03 |
| | 10 % | 2.71 | 1.09 | 0.07 | 1.63 | 0.05 | 2.60 | 1.71 | 0.03 |
| BEL | 0 % | 1.65 ± 0.04 | 0.68 ± 0.03 | 0.03 ± 0.00 | 0.65 ± 0.03 | 0.03 ± 0.00 | 2.26 ± 0.17 | 1.14 ± 0.10 | 0.03 ± 0.00 |
| | 2 % | 1.63 | 0.69 | 0.04 | 0.68 | 0.03 | 2.34 | 1.15 | 0.03 |
| | 10 % | 1.52 | 0.73 | 0.08 | 0.82 | 0.03 | 2.64 | 1.21 | 0.03 |
| 1100 | 100 % | 0.34 ± 0.02 | 1.22 ± 0.05 | 0.45 ± 0.01 | 2.39 ± 0.07 | 0.06 ± 0.00 | 6.00 ± 0.13 | 1.80 ± 0.08 | 0.02 ± 0.00 |

**Table 5-** Definition of the diagnostic pigments used as phytoplankton biomarkers (taxonomic
significance) and associated phytoplankton size class (Uitz et al., 2010).

| Diagnostic Pigments | Abbreviations | Taxonomic Significance | Phytoplankton Size Class |
|---|---|---|---|
| Fucoxanthin | Fuco | Diatoms | microplankton |
| Peridinin | Perid | Dinoflagellates | microplankton |
| 19'-hexanoyloxyfucoxanthin | Hex-fuco | Haptophytes | nanoplankton |
| 19'-butanoyloxyfucoxanthin | But-fuco | Pelagophytes and Haptophytes | nanoplankton |
| Alloxanthin | Allo | Cryptophytes | nanoplankton |
| chlorophyll *b* + divinyl chlorophyll *b* | TChlb | Cyanobacteria, Prochlorophytes | picoplankton |
| Zeaxanthin | Zea | Chlorophytes, Prochlorophytes | picoplankton |




**Figure captions**
**Figure 1-** Bathymetry of the parent and child (grey rectangle) domains interpolated from the GINA
data base with a zoom on the near domain (black rectangle); the oblique white and black lines
represent the large and small sections, respectively, used for numerical simulations.
**Figure 2-** Comparison of temperature and salinity between model outputs and field data at the
OTEC station (a) on June 16[th] 2000 and 2014, respectively and (b) on November 28[th] 2000 and 2013,
respectively.
**Figure 3-** Comparison of mean current direction and horizontal velocity norm between model
outputs from June 2000 and ADCP data from June 2011.
**Figure 4-** Pigment concentrations (from HPLC analysis) at the OTEC site at the DCM (a) and at the
BEL (b), on June 12[th] (D0), 16[th] (D4), 18[th] (D6) 2014 (bars represent the standard deviation).
**Figure 5-** Abundance and biovolume of micro- and part of nano-phytoplankton at the OTEC site on
June 12[th] (D0), 16[th] (D4), 18[th] (D6) 2014, at the DCM (a and c, respectively) and at the BEL (b and d,
respectively) (bars represent the standard deviation).
**Figure 6-** Abundance of pico-phytoplankton at the DCM (a) and at the BEL (b), on June 12[th] (D0),
16[th] (D4), 18[th] (D6) 2014 (bars represent the standard deviation).
**Figure 7-** Specific carbon uptake rate ($h^{-1}$) at the DCM (a) and BEL (b) depths, on June 12[th] (D0), and
18[th] (D6), and in 6 days incubated microcosms (D6), for the three mixing conditions (0 %, 2 % and 10
% of deep seawater additions) (for surrounding waters bars represent the standard deviation for 3
replicates).
**Figure 8-** Diagnostic pigment concentrations in surrounding surface waters on D0 and D6, and in
controls and deep water-enriched (2 % and 10 %) microcosms after 6 days of incubation at the DCM
(bars represent the standard deviation). Similar letters (a, b or c) attributed to 2 or more treatments
indicate no significant differences ($p < 0.05$) between these treatments.
**Figure 9-** Abundance of picophytoplankton in surrounding surface waters on day 0 and 6, and in
controls and deep water-enriched (2 % and 10 %) microcosms after 6 days of incubation at 45 m
depth (a) and 80 m depth (b) (bars represent the standard deviation). Similar letters (a, b or c)
attributed to 2 or more treatments indicate no significant differences ($p < 0.05$) between these
treatments.




**Figure 1**

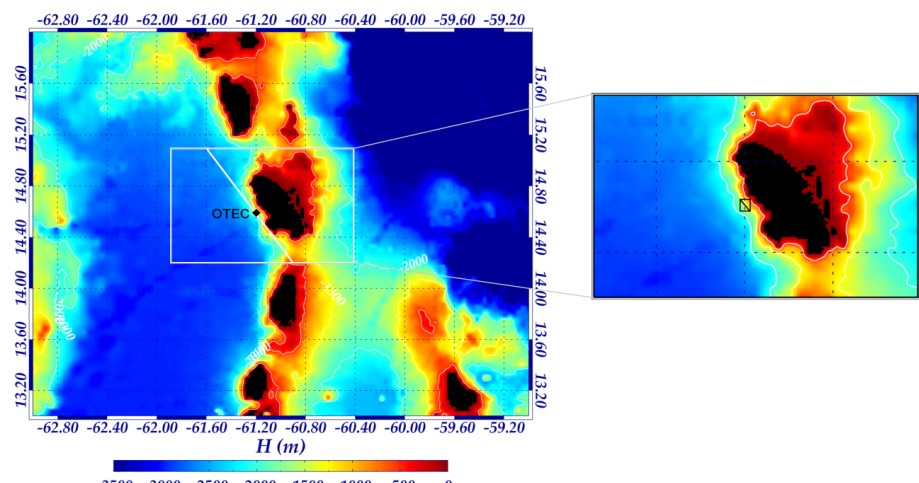

**Figure 2**

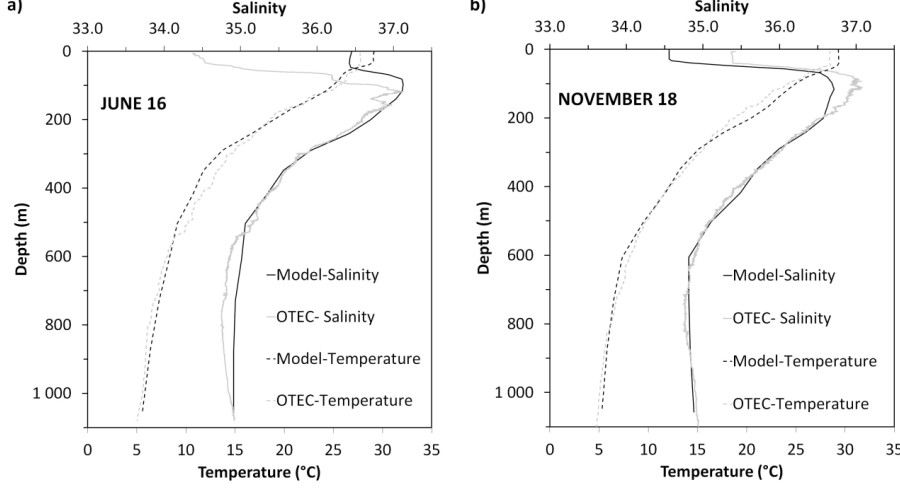






**Figure 3**

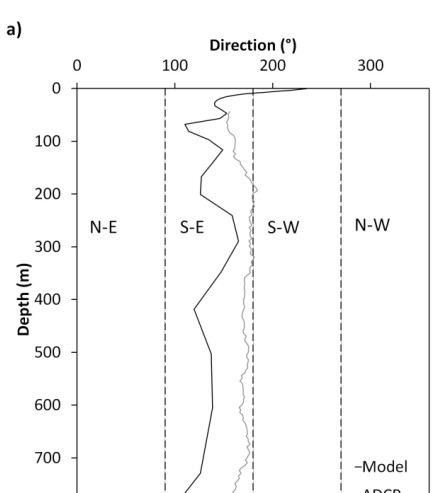
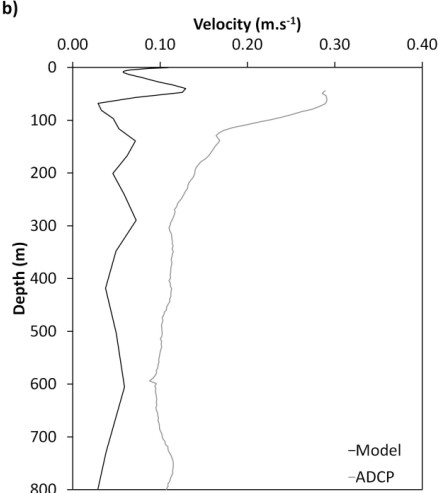

**Figure 4**

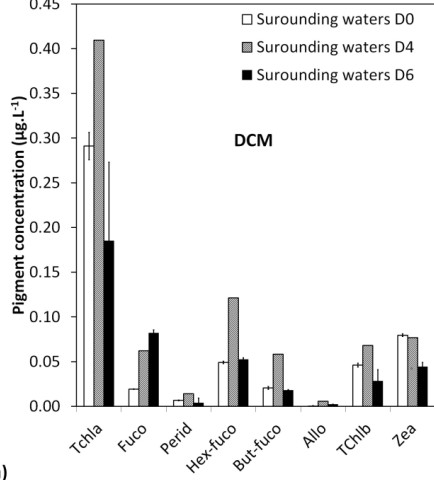
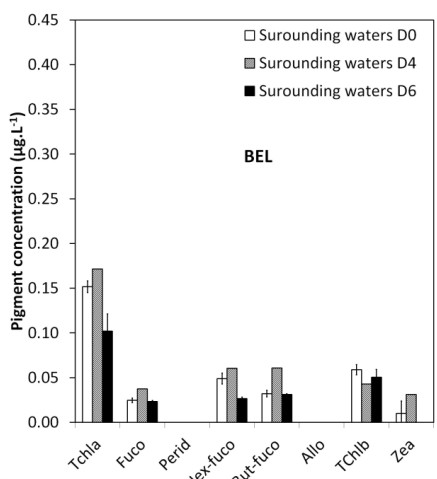






**Figure 5**

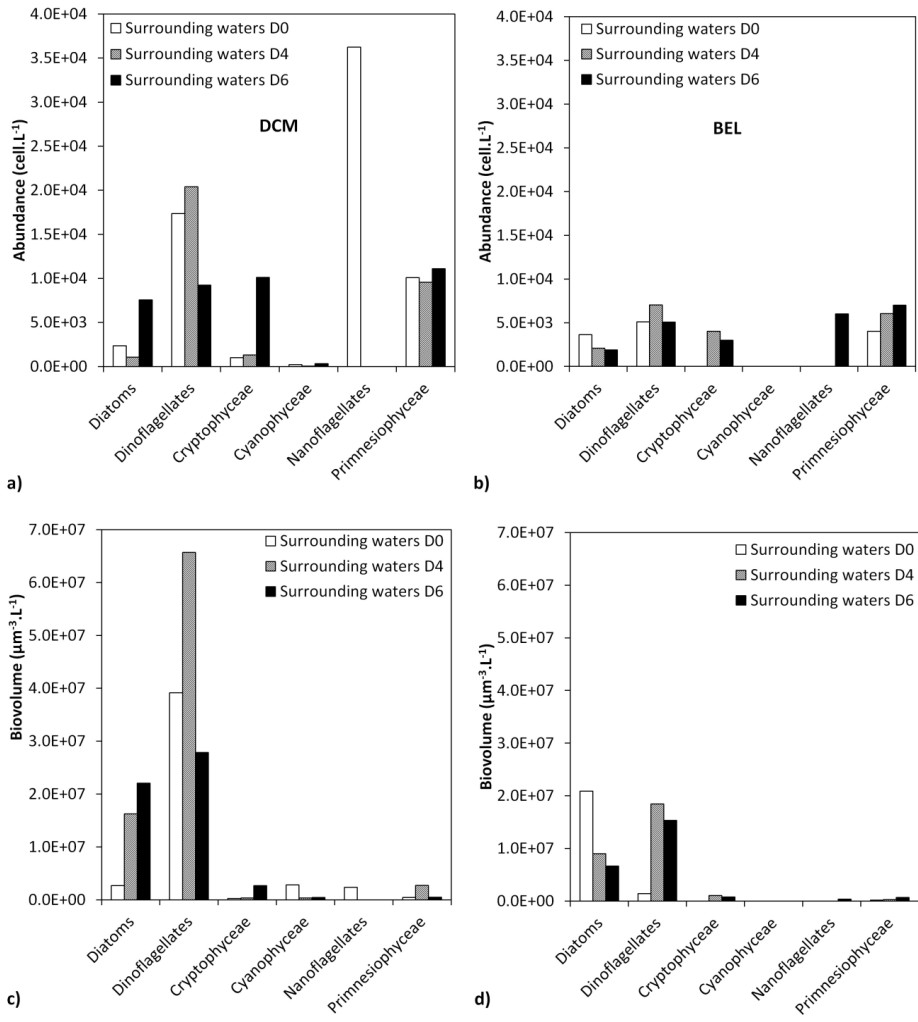






**Figure 6**

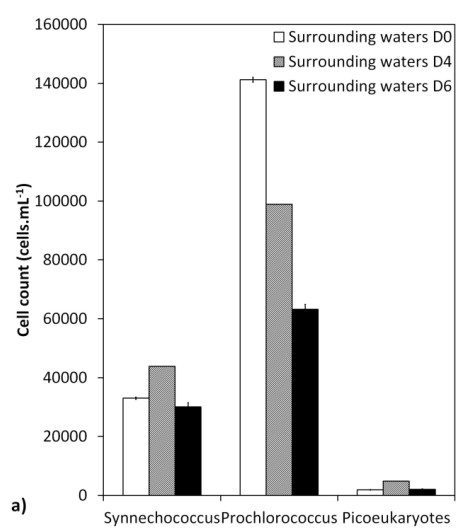
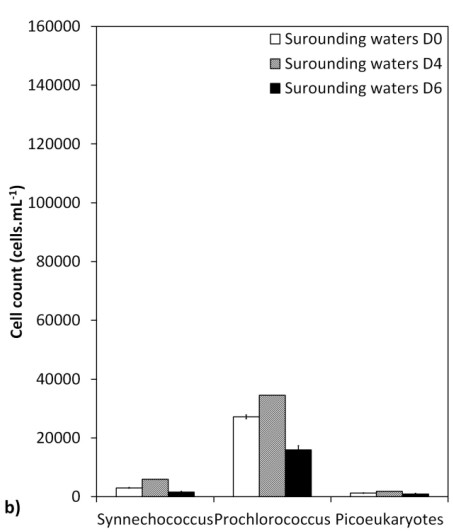

**Figure 7**

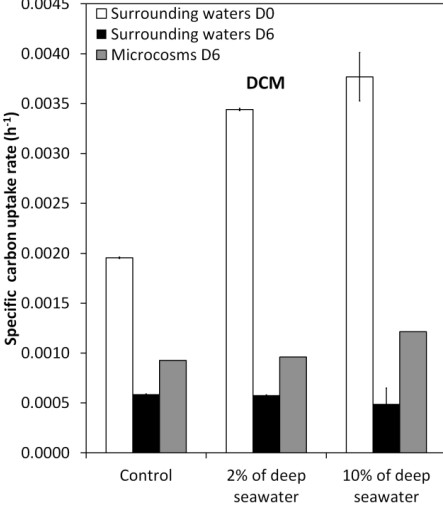
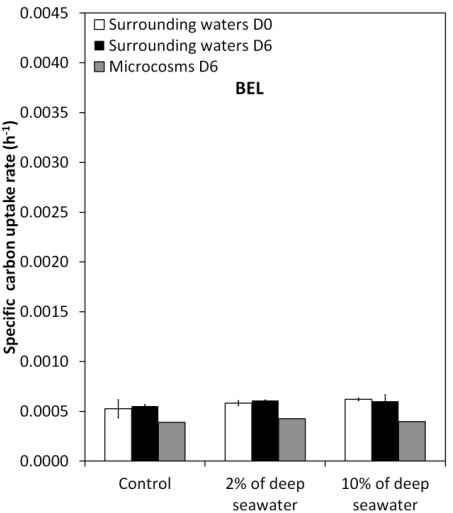




**Figure 8**

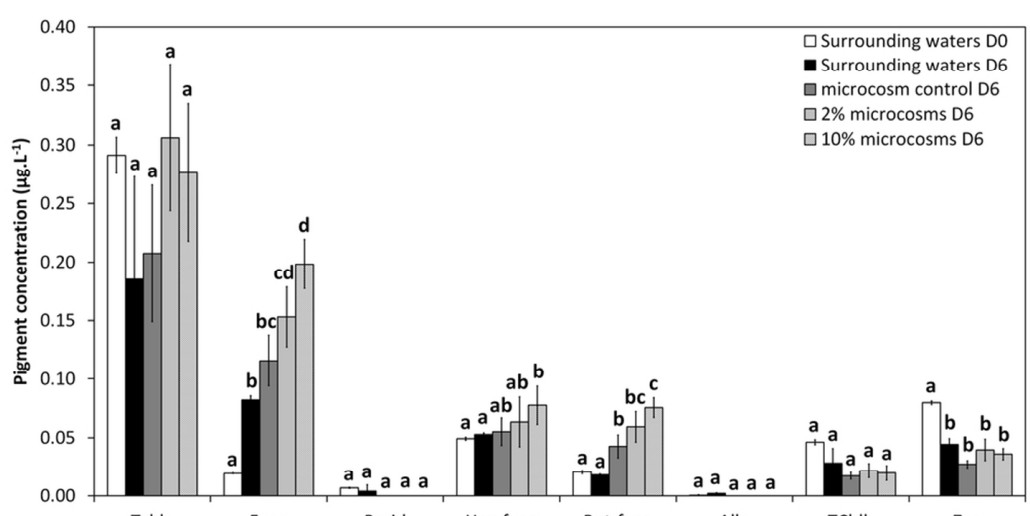

**Figure 9**

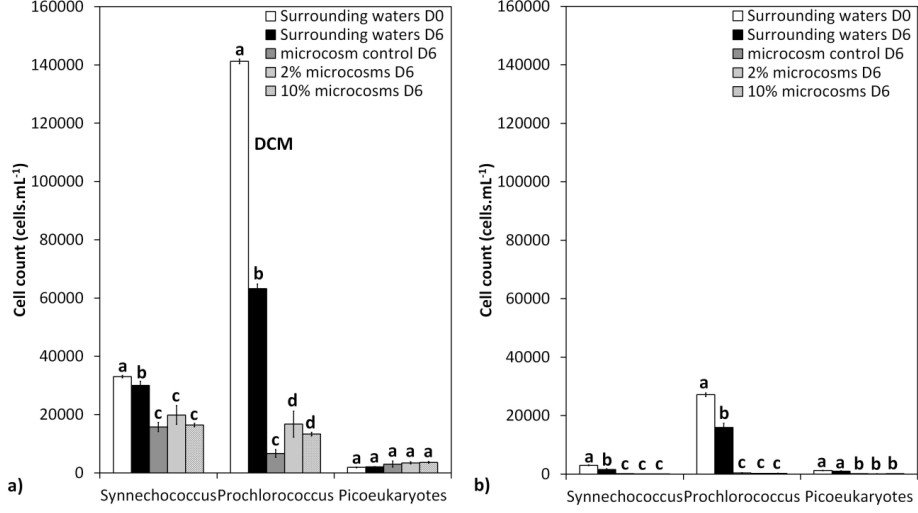

