# Peer review of "Potential effects of deep seawater discharge by an Ocean Thermal Energy"

_Biogeosciences, 2018_

## Referee Comment (RC1) · Anonymous Referee #1 · 2 Sep 2018

Giraud et al examined the potential effects of discharging the cold nutrient-rich deep seawater on the phytoplankton community before the real installation of OTEC pilot plant. Part of the purpose of evaluation is to find a suitable depth where the deep seawater could be discharged without significant effect on the surface phytoplankton community. The effects of discharging seawater is roughly evaluated in two aspects: the thermal effects and their impacts on phytoplankton community. It is a valuable work to evaluate the potential environmental assessment before application of artificial project. However, there is some weaknesses of this paper relate to the validity of the interpretations and implications of the results obtained. 1. The thermal effects of discharging seawater at different depths that a temperature difference of 0.3 °C

and less than 1 km2 on the area was achieved using ROMS- Regional Ocean Model system. The validity of the Model was proven by comparing the modeled temperature, salinity and currents profile with the CTD and ADCP measuring data. However, the comparison between modeled data and in-situ measured data are from different years. Besides, the temperature bias between the modeled data and the in-situ measured data are much higher ($\sim$1.5 °C ) than the simulated thermal effect ïijĹ0.3°CïijĽcaused by OTEC discharging, although author concluded in Line 263 that the modeled physical properties (T, S, Currents) were quite similar to those directly observed at the study site and attributed the differences to inter-annual variability. 2. The purpose of using ROMS- Regional Ocean Model system is to check whether discharging deep seawater would change the phytoplankton community, especially in the surface layer. Since the lowest discharging depth is about 45 m, which is the maximum Chl a depth, the cold deep seawater would mixed with ambient seawater after discharging. whether the mixed water would sink out of thermocline layer is decided by the density. Thus, salinity is also important to check the effect. However, in the result part, the authors did not give the salinity effects caused by discharging, which we believe is an indispensable part.

---

## Author Comment (AC1) · 6 Nov 2018

**Response to Reviewer N°1**

Referee's comments are in italics below.

1. *The thermal effects of discharging seawater at different depths that a temperature difference of 0.3 °C and less than 1 km2 on the area was achieved using ROMS-Regional Ocean Model system. The validity of the Model was proven by comparing the modeled temperature, salinity and currents profile with the CTD and ADCP measuring data. However, the comparison between modeled data and in-situ measured data are from different years. Besides, the temperature bias between the modeled data and the in-situ measured data are much higher (~1.5 °C) than the simulated thermal effect ïjʹL0.3_CïjL'caused by OTEC discharging, although author concluded in Line 263 that the modeled physical properties (T, S, Currents) were quite similar to those directly observed at the study site and attributed the differences to inter-annual variability.*

The only measurements available on the OTEC future location off the western coast of Martinique are those presented in our manuscript, acquired during our field effort (November 2013 and June 2014). At the time of our study, the NCEP-CFSR products did no cover the period of our mesocosm experiments. So we chose to run our model over another period when the atmospheric forcing was available. We chose the 3 years period of 1998-2000, using 1998 and 1999 as a spin-up and the last year 2000 to analyze the thermal structure and circulation field. The thermal structure (depth of thermocline and temperature profile, intermediate and deep waters temperature) is well mimicked despite a slight bias in the very surface due to interannual variability in the atmospheric forcing. The reasonable agreement in the thermal structure allows us to be confident in the estimation of the thermal impact of the OTEC discharge. We agree with the referee about the bias in salinity.

Salinity field data at the OTEC site showed large seasonal variations, with low values in June 2014 (34.6 on the top 50 m) and much higher values in November 2013 (35.5 on the top 50 m). The model run for year 2000 did not fully reproduce these variations. The observations we made at the OTEC station showed that the low salinity observed in June was associated with high Si(OH)$_4$ concentrations. High Si(OH)$_4$ levels in fresher seawater have been already reported in surface waters in the Caribbean Sea and they were attributed to Amazon and Orinoco fresh rivers inputs. Fresh surface waters enriched in Si(OH)$_4$ can be transported from the Amazon and Orinoco rivers towards the Caribbean Sea by the North Brazil Current and the Guiana Current (Muller-Karger et al., 1988, 1995; Osborne et al., 2014, 2015). It is likely that the rivers discharges and thus its inputs in the Caribbean Sea were quite different between 2000 (modeled year) and 2014 (in situ observations), thus explaining the discrepancy between modeled and observed salinities. Meso- and submeso-scale features resulting from the rivers flows could also induce short-term variability in the area and then could explain the observed differences.

2. *The purpose of using ROMS-Regional Ocean Model system is to check whether discharging deep seawater would change the phytoplankton community, especially in the surface layer. Since the lowest discharging depth is about 45 m, which is the maximum Chl a depth, the cold deep seawater would mixed with ambient seawater*

*after discharging. whether the mixed water would sink out of thermocline layer is decided by the density. Thus, salinity is also important to check the effect. However, in the result part, the authors did not give the salinity effects caused by discharging, which we believe is an indispensable part.*

We fully agree with Referee N°1 that density should be looked at. In fact we had looked at it but since we were focusing on the thermal impact on phytoplankton growth, we had decided not to mention it since the impact was also minor. As far as we know it, there are no environmental standards defining threshold levels for density difference that will be induced by an OTEC deep seawater discharge.

The density of water being discharged at 45 m, depth of the deep chlorophyll maximum (DCM), is 27.48 (8°C and salinity of 35). The density of water at 45 m is around 23.72 (temperature of 28°C and salinity of 36.5) so we have a nominal gradient of 3.76 in density. If we consider a modification of the density structure of the top 150 m of the water column of $|\Delta\rho| \geq 0.1$, there is no impact when the discharge occurs at the depth of the DCM. If we consider a lower density difference of 0.05 (absolute value), the area exhibiting a $|\Delta\rho| \geq 0.05$ in the top 150 m is extremely small (< 1.5 km$^2$) in both sections at the depth of the chlorophyll maximum, on an annual average and in June (our experimental period). This represents less than 1.5 % of the nominal density gradient.

---

## Referee Comment (RC2) · Anonymous Referee #2 · 26 Nov 2018

General Comments: In this paper the authors attempt to bring out the hypothetical impact of a hypothetical OTEC plant operating in the offshore region of Caribbean coast of Martinique with the help of numerical model using ROMS and microcosom experiment. Allow me to come to the point straight without any prelude. The manuscript is lost in too lengthy technical details of methodology, most of which is not new and used by several researchers in the field that the authors themselves quote as reference. It is very difficult to read the manuscript, trivial and redundant at times, too many qualitative statements, and contains several contradictions (see specific comments). The manuscript looks more like a technical report than a well articulated scientific paper with sound hypothesis with strong rationale supported with robust data analysis. The

results reported are too general and in my view not substantial (see specific comments) to merit its publication in a peer reviewed scientific journal.

I cannot recommend publication of this manuscript.

Specific Comments:

There are several contradictions and flaws, but I point a few:

1. Title suits more for a technical report.

2. The 17 lines Abstract do not bring out substantial result. The result that is reported in lines 20 to 24 and in lines 24-26 are too general and trivial.

3. Line 29: It is not “bottom” it is “subsurface”

4. Line 37: How narrow?

5. Lines 49-55: Textural material. Unnecessary.

6. Line 56: How low?

7. Lines 57-59: This is a well known fact. Authors need not have to remind the readers. Delete it

8. Lines 61-63: Is this the rationale for the present study? I am not convinced. Authors need to do a better job so that the readers may find it interesting.

9. Lines 91-95: Fjords are very different system in terms of its dynamics as well as its ecosystem. How is this result relevant in the present study which deals with tropical system? The lengthy list of references in this para only serves to increase the volume.

10. Under Materials and Methods

Lines 110-128: There is a miss-match between the period during which the model is forced using dynamic variable and the period of the mesocosm experiments. How justified are the authors to compare the model simulation with mesocosm both being for

different periods without any consideration/justification about the inter-annual variability.

Interestingly, and conveniently in the later part of the manuscript the authors attributes the difference between the observation and model to inter-annual variability (Lines 263-266).

Quite contradictory to this the authors very conveniently concludes in earlier para (lines 241-260) that "modelled physical properties were therefore quite similar to those directly observed"

11. Under Results

Lines 241-260: The model simulated temperature profile has a warm bias of 1.5oC in the upper 50m and cold bias of 1.5oC between 300 and 500m in June. A similar over estimation is also seen in the model salinity in the upper 60m which was as large as 2 units. Similarly, in November also the temp showed a warm bias, however, the salinity was under estimated. The problem of the model is not limited to T and S, the sub-surface current speed also showed large deviation from that of ADCP (lines 261-262).

In spite of such deviations between the model and observation authors conclude that modelled physical properties are quite similar to observation (lines 263-264) which is unacceptable and I disagree. Authors need to do a better job.

12. Under Conclusion Lines 537-555: What is the important "take-home message"? "The phytoplankton community and its production could be impacted by a large deep seawater input"? (lines 548-549)

13. Other major concern is the utility of the present study which is limited to such a small spatio-temporal scale and its impact/relevance in the context of the open ocean processes such as mixing aided by the air-sea flux variability driven by the winds.

---

## Author Comment (AC2) · 5 Dec 2018

**Response to Reviewer N°1**
Referee's comments are in italics below.

*1. The thermal effects of discharging seawater at different depths that a temperature difference of 0.3 °C and less than 1 km² on the area was achieved using ROMS- Regional Ocean Model system. The validity of the Model was proven by comparing the modeled temperature, salinity and currents profile with the CTD and ADCP measuring data. However, the comparison between modeled data and in-situ measured data are from different years. Besides, the temperature bias between the modeled data and the in-situ measured data are much higher (~1.5 °C) than the simulated thermal effect ïïj´L0.3_CïïjL'caused by OTEC discharging, although author concluded in Line 263 that the modeled physical properties (T, S, Currents) were quite similar to those directly observed at the study site and attributed the differences to inter-annual variability.*

The only measurements available on the OTEC future location off the western coast of Martinique are those presented in our manuscript, acquired during our field effort (November 2013 and June 2014). At the time of our study, the NCEP-CFSR products did no cover the period of our mesocosm experiments. So, we chose to run our model over another period when the atmospheric forcing was available. We chose the 3 years period of 1998-2000, using 1998 and 1999 as a spin-up and the last year 2000 to analyze the thermal structure and circulation field. The thermal structure (depth of thermocline and temperature profile, intermediate and deep waters temperature) is well mimicked despite a slight bias in the very surface due to interannual variability in the atmospheric forcing. The reasonable agreement in the thermal structure allows us to be confident in the estimation of the thermal impact of the OTEC discharge. We agree with the referee about the bias in salinity.

Salinity field data at the OTEC site showed large seasonal variations, with low values in June 2014 (34.6 on the top 50 m) and much higher values in November 2013 (35.5 on the top 50 m). The model run for year 2000 did not fully reproduce these variations. The observations we made at the OTEC station showed that the low salinity observed in June was associated with high $Si(OH)_4$ concentrations. High $Si(OH)_4$ levels in fresher seawater have been already reported in surface waters in the Caribbean Sea and they were attributed to Amazon and Orinoco fresh rivers inputs. Fresh surface waters enriched in $Si(OH)_4$ can be transported from the Amazon and Orinoco rivers towards the Caribbean Sea by the North Brazil Current and the Guiana Current (Muller-Karger et al., 1988, 1995; Osborne et al., 2014, 2015). It is likely that the rivers discharges and thus its inputs in the Caribbean Sea were quite different between 2000 (modeled year) and 2014 (in situ observations), thus explaining the discrepancy between modeled and observed salinities. Meso- and submesoscale features resulting from the rivers flows could also induce short-term variability in the area and then could explain the observed differences.

We propose to reformulate lines 263-265 as follows:

The fit between modeled physical properties (temperature, salinity, currents) and those directly observed at the study site is not perfect due to interannual variability in atmospheric forcing and freshwater inputs by the major rivers but also to meso- and submesoscale variability present in the region. However, the estimation of the thermal impact of the OTEC discharge which is our major objective here can be considered with confidence since the thermocline and halocline depths, key proxies for oceanic mixing, are well mimicked in June and November.

*2. The purpose of using ROMS-Regional Ocean Model system is to check whether discharging deep seawater would change the phytoplankton community, especially in the surface layer. Since the lowest discharging depth is about 45 m, which is the maximum Chl a depth, the cold deep seawater would mixed with ambient seawater after discharging. whether the mixed water would sink out of thermocline layer is decided by the density. Thus, salinity is also important to check the effect. However, in the result part, the authors did not give the salinity effects caused by discharging, which we believe is an indispensable part.*

We fully agree with Referee N°1 that density should be looked at. In fact, we had looked at it but since we were focusing on the thermal impact on phytoplankton growth, we had decided not to mention it since the impact was also minor. As far as we know it, there are no environmental standards defining threshold levels for density difference that will be induced by an OTEC deep seawater discharge.

The density of water being discharged at 45 m, depth of the deep chlorophyll maximum (DCM), is 27.48 (8°C and salinity of 35). The density of water at 45 m is around 23.72 (temperature of 28°C and salinity of 36.5) so we have a nominal gradient of 3.76 in density. If we consider a modification of the density structure of the top 150 m of the water column of $|\Delta\rho| \geq 0.1$, there is no impact when the discharge occurs at the depth of the DCM. If we consider a lower density difference of 0.05 (absolute value), the area exhibiting a $|\Delta\rho| \geq 0.05$ in the top 150 m is extremely small (< 1.5 km²) in both sections at the depth of the chlorophyll maximum, on an annual average and in June (our experimental period). This represents less than 1.5 % of the nominal density gradient.

We propose to insert the following section after line 288 (just before Section 3.2):

The density of water being discharged at 45 m, depth of the deep chlorophyll maximum (DCM), is 27.48 (8°C and salinity of 35). The density of water at 45 m is around 23.72 (temperature of 28°C and salinity of 36.5) so the nominal density gradient is of 3.76. If one considers a modification of the density structure of the top 150 m of the water column of $|\Delta\rho| \geq 0.1$, there is no impact when the discharge occurs at the depth of the DCM. If one considers a lower density difference of 0.05 (absolute value), the area exhibiting a $|\Delta\rho| \geq 0.05$ in the top 150 m is extremely small (< 1.5 km²) in both sections at the depth of the chlorophyll maximum, on an annual average and in June (our experimental period). As far as we know, there are no environmental standards defining threshold levels for density difference that will be induced by an OTEC deep seawater discharge. This represents less than 1.5 % of the nominal density gradient so as for the thermal impact the impact is estimated to be minor.

---

## Author Comment (AC3) · 5 Dec 2018

**Response to Reviewer N°2**
Referee's comments are in italics below.

*General Comments: In this paper the authors attempt to bring out the hypothetical impact of a hypothetical OTEC plant operating in the offshore region of Caribbean coast of Martinique with the help of numerical model using ROMS and microcosom experiment. Allow me to come to the point straight without any prelude. The manuscript is lost in too lengthy technical details of methodology, most of which is not new and used by several researchers in the field that the authors themselves quote as reference. It is very difficult to read the manuscript, trivial and redundant at times, too many qualitative statements, and contains several contradictions (see specific comments). The manuscript looks more like a technical report than a well articulated scientific paper with sound hypothesis with strong rationale supported with robust data analysis. The results reported are too general and in my view not substantial (see specific comments) to merit its publication in a peer reviewed scientific journal. I cannot recommend publication of this manuscript.*

We are quite shocked by the contemptuous tone of these few lines. Why twice "hypothetical" is used in the same first sentence? Is referee N°2 insinuating we are lying about this OTEC plant planning off Martinique? This project was funded by France Energies Marines which is the Institute for energy transition dedicated to renewable energies located in Brest, France. This work was conducted in the context of the ecological transition to renewable blue energies. No environmental standard is currently available worldwide on the thermal energies of the seas. Our study is therefore an essential and unique step before the commissioning of such plants, in addition to bring original results on the impact of surface water fertilization in the oligotrophic domain. We find it unfortunate and worrying that the reviewer did not understand the importance of this scientific study especially in the context of the development of renewable energies. We think this prelude is uselessly aggressive.

We do not think there is unnecessary detail on techniques and methods, since our experimental study is based on many measured biological, biogeochemical and physicochemical parameters, in addition to a modeling part. Besides, we are astonished by these comments of Referee N°2 and by finding that our article looks like a technical report. Maybe the referee is not used to conducting experimental studies and/or he/she missed that our study evaluates the potential impacts of deep seawater discharge in the subsurface oligotrophic waters? Furthermore, the data have been analyzed using statistical analyses at two levels (Kruskal-Wallis test and Mann-Whitney test when necessary) to provide robust results.

There is absolutely no contradiction in our analysis. Indeed, we used the atmospheric forcing which was available to us at the time of the study and showed that the representation of the thermocline and halocline depths, key proxies for oceanic mixing and for estimating the thermal impact of the OTEC discharge, is very well mimicked over the 2 months of the mesocosm experiments. We recognized however the existence of temperature and salinity biases and we discussed the potential underlying reasons for this, being interannual variability in atmospheric forcing and freshwater inputs by the major rivers but also to meso- and submesoscale variability present in the region.

*Specific Comments:*
*There are several contradictions and flaws, but I point a few:*

*1. Title suits more for a technical report.*

We do not think that mentioning the term OTEC in the title preludes to a technical report. Indeed, it sets the framework for this study in the context of renewable blue energies, since the aim of our study

is well mentioned in the title ("potential effects of deep seawater discharge... on the marine microorganisms in oligotrophic waters").

*2. The 17 lines Abstract do not bring out substantial result. The result that is reported in lines 20 to 24 and in lines 24-26 are too general and trivial.*

Descriptive and trivial are two different things, and trivial is inappropriate regarding our study.

We propose to add quantitative results and details in the abstract, as follows:

Line 23- … with a development of diatoms (increase by 71% of fucoxanthin pigment concentration after 6 days of incubation, as compared to the controls without deep seawater addition) and haptophytes (increase by 77% of the 19'-butanoyloxyfucanxanthin pigment concentration at day 6 as compared to the controls)

Line 24- … limited change of the phytoplankton community (higher Prochlorococcus abundance than the controls at day 6, but without significant shift of the assemblage and primary production)

Line 25-…significantly modify the phytoplankton assemblage with a shift from pico-phytoplankton toward micro-phytoplankton …

*3. Line 29: It is not "bottom" it is "subsurface"*

No: the temperature gradient used to operate the plant is the one between the "warm" surface water and the "cold" bottom water (a gradient of 20°C is needed for the OTEC exploitation as indicated in Line 33).

*4. Line 37: How narrow?*

The plateau is narrow enough so that the depth reaches 1300 meters at only 5 km from the coast. Figure 1 is already showing the bathymetry.

*5. Lines 49-55: Textural material. Unnecessary.*

We do not think that describing the functioning of a natural upwelling is a material of texture since we juxtapose it to the artificial upwelling that will be generated by the plant.

*6. Line 56: How low?*

As shown in Table 3 and indicated at Line 296, nitrate and phosphate concentrations are below the detection limit (< 0.02 µm) in the euphotic layer. We will add this value at Line 56.

*7. Lines 57-59: This is a well known fact. Authors need not have to remind the readers. Delete it*

We do not agree. We think it is important to precise the phytoplankton assemblage differences between a natural upwelling and oligotrophic waters. Indeed, despite very different ambient assemblages, the 10% deep water addition to oligotrophic waters is leading to significant shift in the assemblage notably toward diatoms like in natural upwelling.

*8. Lines 61-63: Is this the rationale for the present study? I am not convinced. Authors need to do a better job so that the readers may find it interesting.*

The rationale is described throughout the introduction, not only in those lines (for instance, see also lines 42-43, Lines 66-67).

For the clarity of the rationale, we propose to merge Lines 66-67 with Lines 61-63 and to add some precisions, as follows:

Due to these important differences in biogeochemical functioning and environmental microbiology, it is thus of critical interest to investigate the potential effects of the deep seawater discharge of the planned OTEC plant on the phytoplankton community off Martinique. Furthermore, it is crucial to provide a depth where the deep seawater could be discharged without significant effect on the surface layer where phytoplankton is the most abundant. Indeed, no environmental standards on the deep seawater discharge effects are available yet, while transitional blue energies such as OTEC plants will likely expand in the near future.

*9. Lines 91-95: Fjords are very different system in terms of its dynamics as well as its ecosystem. How is this result relevant in the present study which deals with tropical system? The lengthy list of references in this para only serves to increase the volume.*

Indeed, fjords are different compared to oligotrophic systems although both can present (seasonally or permanently) low nutrients levels. So, we find relevant to mention the experiment of artificial upwelling that has been conducted in the Norwegian fjord.

We do not need to increase the volume of the text (for which we will have to pay), but we just want to be exhaustive on the experiments conducted in the field.

*10. Under Materials and Methods*
*Lines 110-128: There is a miss-match between the period during which the model is forced using dynamic variable and the period of the mesocosm experiments. How justified are the authors to compare the model simulation with mesocosm both being for different periods without any consideration/justification about the inter-annual variability. Interestingly, and conveniently in the later part of the manuscript the authors attributes the difference between the observation and model to inter-annual variability (Lines 263-266). Quite contradictory to this the authors very conveniently concludes in earlier para (lines 241-260) that "modelled physical properties were therefore quite similar to those directly observed".*
*11. Under Results*
*Lines 241-260: The model simulated temperature profile has a warm bias of 1.5°C in the upper 50m and cold bias of 1.5°C between 300 and 500m in June. A similar over estimation is also seen in the model salinity in the upper 60m which was as large as 2 units. Similarly, in November also the temp showed a warm bias, however, the salinity was under estimated. The problem of the model is not limited to T and S, the sub-surface current speed also showed large deviation from that of ADCP (lines 261-262). In spite of such deviations between the model and observation authors conclude that modelled physical properties are quite similar to observation (lines 263-264) which is unacceptable and I disagree. Authors need to do a better job.*

We indeed made the choice of choosing monthly forcing from reanalyses. The monthly Climate Forecast System Reanalysis (NCEP-CFSR) for wind stress, heat and freshwater fluxes does not provide yet this forcing for the years 2013 and 2014 where the mesocosm experiments have been carried out.

Even now, the atmospheric fields are only available until end of December 2011 so do not cover 2013 and 2014. Needless to say, of course we are aware of interannual variability in the forcing and in the ocean's response to this variable forcing. We chose the 3 years period of 1998-2000, using 1998 and 1999 as a spin-up and the last year 2000 to analyze the thermal structure and circulation field. The goal here is to try to get a reasonable depiction of water masses in presence in the region, not a perfect match between observed and modelled temperature and salinity profiles, since we knew for interannual variability reasons, this will not be possible. The reasonable description of the thermocline and halocline depths and water masses we obtained allows one to be confident in the estimation of the thermal impact of the OTEC discharge which is our major objective here.

The thermal structure (depth of thermocline and temperature profile, intermediate and deep waters temperature) is well mimicked despite a slight bias in the very surface due to interannual variability in the atmospheric forcing. Salinity field data at the OTEC site showed large seasonal variations, with low values in June 2014 (34.6 on the top 50 m) and much higher values in November 2013 (35.5 on the top 50 m). The model run for year 2000 did not fully reproduce these variations. The observations we made at the OTEC station showed that the low salinity observed in June was associated with high $Si(OH)_4$ concentrations. High $Si(OH)_4$ levels in fresher seawater have been already reported in surface waters in the Caribbean Sea and they were attributed to Amazon and Orinoco fresh rivers inputs. Fresh surface waters enriched in $Si(OH)_4$ can be transported from the Amazon and Orinoco rivers towards the Caribbean Sea by the North Brazil Current and the Guiana Current (Muller-Karger et al., 1988, 1995; Osborne et al., 2014, 2015). The rivers discharges and thus its inputs in the Caribbean Sea were quite different between 2000 (modeled year) and 2014 (in situ observations), thus explaining the discrepancy between modeled and observed salinities. Meso- and submesoscale features resulting from the rivers flows could also induce short-term variability in the area and then could explain the observed differences.

We propose to reformulate lines 263-265 as follows:

The fit between modeled physical properties (temperature, salinity, currents) and those directly observed at the study site is not perfect due to interannual variability in atmospheric forcing and freshwater inputs by the major rivers but also to meso- and submesoscale variability present in the region. However, the estimation of the thermal impact of the OTEC discharge which is our major objective here can be considered with confidence since the thermocline and halocline depths are very well mimicked in June and November.

*12. Under Conclusion Lines 537-555: What is the important "take-home message"? "The phytoplankton community and its production could be impacted by a large deep seawater input"? (lines 548-549)*

Conclusive remarks are given on the thermal effect of deep seawater discharge (Lines 539-547), on the impact on microorganisms (Lines 548-555) and on the perspectives of this work (Lines 556-561).

We can further specify another conclusive result of our experiments (after Line 555), as following:

Since the lower impact on the phytoplankton assemblage was obtained at BEL, this depth can be recommended for the discharge the deep seawater to exploit the OTEC plant.

*13. Other major concern is the utility of the present study which is limited to such a small spatio-temporal scale and its impact/relevance in the context of the open ocean processes such as mixing aided by the air-sea flux variability driven by the winds.*

As indicated in Lines 529-530 and in Lines 556-561, our *in situ* experiments did not allow drawing a scenario over the long term impact of the OTEC plant exploitation. But our experiments provide the essential basis to evaluate the depth where the deep seawater could be discharged with a minimum effect on the phytoplankton assemblage. For comparison, it is like the Fe experiments that were first conducted in batch incubations in High Nutrients Low Chlorophyll domains to test the Iron Hypothesis of John Martin (1990), until the artificial Fe fertilizations were conducted years later. Long term monitoring will be necessary once the plant will be operated as indicated in Lines 558-561.